# Improving patient-centred counselling skills among lay healthcare workers in South Africa using the Thusa-Thuso motivational interviewing training and support program

Idah Mokhele[1], Tembeka Sineke[1], Marnie Vujovic[2], Robert A. C. Ruiter[3], Jacqui Miot[1], Dorina Onoya[1] *

1 Health Economics and Epidemiology Research Office, Faculty of Health Sciences, University of the Witwatersrand, Johannesburg, South Africa, 2 ANOVA Health Institute, Johannesburg, South Africa, 3 Department of Work and Social Psychology, Maastricht University, Maastricht, the Netherlands

* donoya@heroza.org

**Data Availability Statement:** All relevant qualitative data are within the paper. The quantitative data that support the findings of this study are available in

## Abstract

We developed a motivational interviewing (MI) counselling training and support program for lay counsellors in South Africa–branded "Thusa-Thuso—*helping you help*", commonly referred to as Thusa-Thuso. We present the results of a pilot study to determine the program's impact on MI technical skills and qualitatively assess the feasibility of a training-of-trainers (TOT) scale-up strategy among counselling staff of non-governmental (NGO) support partners of the human immunodeficiency virus (HIV) treatment program in South Africa. We enrolled adult ($\geq$ 18 years) lay counsellors from ten primary healthcare clinics in Johannesburg (South Africa) selected to participate in the Thusa-Thuso training and support program. Counsellors attended the ten-day baseline and quarterly refresher training over 12 months (October 2018-October 2019). Each counsellor submitted two audio recordings of mock counselling sessions held during the ten-day baseline training and two additional recordings of sessions with consenting patients after each quarterly contact session. We reviewed the recordings using the MI treatment integrity (MITI) coding system to determine MI technical (cultivating change talk and softening sustain talk) and relational (empathy and partnership) competency scores before and after training. After 12 months of support with pilot site counsellors, we were asked to scale up the training to NGO partner team trainers in a once-off five-day Training of trainers (TOT) format (n = 127 trainees from November 2020 to January 2021). We report TOT training experiences from focus group discussions (n = 42) conducted six months after the TOT sessions. Of the 25 enrolled lay counsellors from participating facilities, 10 completed the 12-month Thusa-Thuso program. Attrition over the 12 months was caused by death (n = 3), site exclusion/resignations (n = 10), and absence (n = 2). MI competencies improved as follows: the technical skills score increased from a mean of 2.5 (standard deviation (SD): 0.8) to 3.1 (SD: 0.5), with a mean difference of 0.6 (95% confidence interval (CI): 0.04, 0.9). The MI relational skills score improved from a mean of 3.20 (SD: 0.7) to 3.5 (SD: 0.6), with a mean difference of 0.3 (95% CI: -0.3, 8.5). End-point qualitative data from the counsellors highlighted the value of identifying and

the public data repository platform "Figshare" and can be easily accessed using the link: https://figshare.com/s/5344ffeab856882fd9fb.

**Funding:** This study has been made possible by the generous support of the American People and the President's Emergency Plan for AIDS Relief (PEPFAR) through the United States Agency for International Development (USAID). DO, IM, JM, and TS were supported through funding from USAID from the Cooperative Agreements 674-A-12-00029 and 72067419CA00004 to the Health Economics and Epidemiology Research Office. MV was funded under the Cooperative Agreement 674-A-12-00020 to Right to Care. RR did not receive specific funding for this work. The contents are the authors' responsibility and do not necessarily reflect the views of PEPFAR, USAID, or the United States government. The funders had no role in study design, data collection and analysis, decision to publish, or preparation of the manuscript.

**Competing interests:** The authors have declared that no competing interests exist.

addressing specific skill deficiencies and the importance of counsellors being able to self-monitor skill development using the MITI review process. Participants appreciated the ongoing support to clarify practical MI applications. The TOT program tools were valuable for ongoing on-the-job development and monitoring of quality counselling skills. However, the MITI review process was perceived to be too involved for large-scale application and was adapted into a scoring form to document sit-in mentoring sessions. The Thusa-Thuso MI intervention can improve counsellor motivation and skills over time. In addition, the program can be scaled up using an adapted TOT process supplemented with fidelity assessment tools, which are valuable for skills development and ongoing maintenance. However, further studies are needed to determine the effect of the Thusa-Thuso program on patient ART adherence and retention in care.

**Trial registration**: Pan African Clinical Trials Registry No: PACTR202212796722256 (12 December 2022).

## Introduction

South Africa's implementation of the Universal test and treat (UTT) policy and same-day initiation (SDI) strategy has resulted in 5.5 million HIV-positive patients initiating antiretroviral therapy (ART) by 2021 [1–3]. These policies offer immediate treatment upon diagnosis, leading to improved access and earlier initiation of ART [1]. With the SDI strategy, 80% to over 90% of HIV-eligible patients initiate ART within 30 days [4, 5]. However, 20% of patients still delay starting treatment six months after diagnosis despite the availability of early initiation [6].

Challenges in patient retention persist under the UTT and SDI policy in South Africa, leading to a shift in patient attrition from the pre-ART phase after diagnosis to the period after ART initiation [6]. Factors such as denial of HIV diagnosis, fears of social isolation, and concerns about medication side effects can hinder sustained ART adherence [7, 8]. Adequate patient support is crucial for long-term retention in care [9]. The clinic environment, support networks, and self-efficacy for ART adherence are all facilitators of patients remaining in care for over ten years [10]. However, implementing universal and early ART policies has placed an additional burden on the healthcare system, gradually reducing the quality of psychosocial support for HIV-positive clients at primary healthcare (PHC) clinics [10, 11].

Patient-centred care focuses on treating patients as individuals, considering their values and preferences in healthcare delivery [12]. However, policy directives promoting patient-centred care in the public health sector in South Africa lack clearly defined methodologies for training health workers in their implementation [13]. The critical shortage of health professionals in South Africa, especially in rural areas, also poses a major obstacle to achieving patient-centred care and universal access to HIV treatment [14–16]. Through task-shifting, lay counsellors have become pivotal in HIV testing, counselling, and providing psychosocial and support services in PHC clinics [17–19]. However, due to minimal training and support, they often have limited basic counselling skills, limiting their effectiveness in balancing the demands of providing HIV testing services and quality patient-centred counselling [20, 21].

With increasing emphasis on patient-centred counselling, motivational interviewing (MI) is being incorporated into models of care to help patients identify and address obstacles they may face in accessing and adhering to ART [7, 22–24]. MI requires the application of both

relational and technical skills. Relational skills encompass active listening, empathy, and building rapport. These skills are crucial for establishing a collaborative and empathetic relationship between the counsellor and client [25, 26]. Technical skills involve specific techniques like open-ended questioning, reflective listening, summarizing, affirming, and eliciting the patient's motivation for change, enabling the counsellor to explore ambivalence, enhance motivation, and facilitate behaviour change [25, 26]. MI relational and technical skills have been shown to positively impact clients' contemplation and reflection on behaviour change [25, 27, 28].

However, MI is typically delivered by trained therapists and skilled healthcare professionals [27, 29–31]. Limited evidence exists for training lay counsellors in MI in PHC settings in sub-Saharan Africa and the use of MI counselling as a strategy for promoting ART uptake and retention. When applied in risk reduction counselling in South Africa, ineffective counsellor training and the lack of ongoing mentoring support have resulted in limited impact [32]. Additionally, although culturally adapted MI counselling has demonstrated therapeutic benefits across diverse populations [33–35], efforts to adapt MI to sub-Saharan African contexts, particularly involving lay health workers, have been minimal [29, 36, 37].

In the recent PAED-LINK trial, lay personnel were trained in MI counselling using online training tools for the brief negotiated interviewing (BNI) intervention [38]. The BNI is a structured MI counselling approach that addresses risky behaviours and promotes behaviour change in emergency healthcare settings [39]. The PAED-LINK trial demonstrated high acceptability for MI counselling methods. Still, it highlighted the need for contextual adaptations, fidelity assessment and mentoring support to facilitate learning and retention of MI skills [38]. To address these deficiencies and establish an approach for training lay counsellors in patient-centred counselling, we used the Intervention Mapping protocol, a framework for developing evidence-based health promotion programs [40]. We incorporated the BNI training tools [39] and partnered with counselling skills trainers to develop a training program collaboratively. The intervention we developed, titled "Thusa-Thuso—*helping you help*", commonly referred to as Thusa-Thuso, is a contextually relevant MI training program for continued MI skill development and support for lay counsellors in South Africa [41].

We report findings from a pilot study to determine the effect of the Thusa-Thuso program on lay counsellors' MI technical and relational skills. In addition, we describe the results of a qualitative assessment of the feasibility of a training-of-trainers (TOT) scale-up model among non-governmental (NGO) partners of the HIV treatment program in South Africa.

## Materials and methods

### Overview of the study population

We conducted a pilot, mixed-method study comprising lay counsellors from ten public PHC clinics randomly assigned to receive the MI training intervention as part of a planned randomized controlled trial in Johannesburg, South Africa (trial registration number: PACTR202212796722256). These PHC clinics currently provide HIV counselling and testing services as well as manage HIV-positive patients on ART. All lay counsellors working at the selected PHC clinics were approached to participate in the study. Drawing from our prior experiences with the ten PHC clinics in other research studies, where we consistently observed an average of three counsellors working per facility, we aimed to recruit 30 counsellors for the current study.

Counsellors were referred to the study staff by facility managers. They were eligible to participate in the study if they regularly worked at the selected facility and were willing to

participate in the 12-month Thusa-Thuso training and support program and willing to participate in study interviews at baseline, post-training, and after 12 months.

## The Thusa-Thuso MI training intervention

The Thusa-Thuso MI training program aims to improve general HIV counselling skills, build sustained MI skills, and ultimately improve ART uptake and retention in care among persons living with HIV (PLHIV). The program consists of a ten-day baseline training followed by two-day quarterly refresher training and debriefing sessions for the 12-month training duration [41]. Training methods applied in the program include didactic training sessions, observation learning through skills demonstration videos that model "good" and "bad" MI counselling and opportunities for skills practice with qualitative training facilitator and peer feedback during onsite baseline and quarterly refresher training sessions to cultivate a supportive learning environment. The training program includes formal feedback via the motivational interviewing treatment integrity (MITI) system, a validated system used to assess and provide feedback on the fidelity and quality of MI interventions to improve proficiency in MI counselling [42, 43].

The HIV treatment readiness framework, a process model for ART readiness guided the training, provided in S1 Fig and published elsewhere [41], underscoring that readiness is based on correct knowledge, perceived importance, MI skills, and readiness for quality counselling [41]. It is based on the information, motivational and behavioural (IMB) skills model, which emphasizes accurate information, motivation, and essential skills in shaping health behaviours [44]. It also incorporates change theory, which explores transitions in behaviour adoption stages, emphasizing readiness and motivation for change [45]. The theory of planned behaviour is also integrated, which suggests that attitudes, norms, and control guide human behaviour and affect intentions and actions [46].

## Data collection

**Survey data collection.**   Trained study staff screened potential eligible lay counsellors and obtained written informed consent among those eligible in their preferred language (English, Sotho, or Zulu). Study enrolment took place from April to October 2018. Counsellors participated in the Thusa-Thuso MI training and support program from October 2018 to October 2019, during which participants were encouraged to use the MI approach in their routine HIV testing counselling sessions and provided with support for its implementation.

We interviewed counsellors at study enrolment using an interviewer-administered baseline questionnaire. Study questionnaires were written in English and administered in the participant's preferred language (English, Sotho, or Zulu). Data collected included socio-demographic information, training background, as well as personal and work-related contextual factors.

**Qualitative data collection.**   We conducted focus group discussions (FGD) among lay counsellors from the study sites after completing the Thusa-Thuso training pilot (at 12 months). FGDs were conducted by trained interviewers to gather information regarding lay counsellor experiences of the training program, including how the training impacted their counselling approach, interactions with clients, and motivation related to their responsibilities in their work environment. The FGD lasted approximately 60 minutes and was conducted outside the lay counsellors' workplaces.

After completion of the pilot project, we were requested by The United States Agency for International Development (USAID) to conduct a once-off training-of-trainers (TOT) scale-up training among non-governmental organizations (NGO) partners of the HIV treatment program in South Africa. The TOTs took place from November 2020 to January 2021 with 127

**Table 1. Overview of study participants for the focus group discussions (FGDs) for scalability and process evaluation.**

| Type of provider | Total |
| --- | --- |
| Lay counsellor trainers/supervisors | 3 |
| Project Coordinators | 3 |
| Social auxiliary workers | 6 |
| Enrolled nursing assistant | 1 |
| Technical advisors: Psychosocial support | 2 |
| Lay HIV counsellor | 15 |
| Peer educators | 2 |
| Community Systems Technical Officers (CSTO) | 10 |
| **Total** | **42** |

trainees. Each training session consisted of 20–25 trainees, facilitated by four facilitators. The TOT training spanned five days; it consisted of three-day interactive Zoom/in-person sessions, a practical in-clinic counselling day with live mentoring support, and a final day for reviewing concepts and experiences during the practice day. The TOT was designed to impart MI counselling and training skills, as these skills are required to roll out the training to their teams.

After the TOT training, we conducted six-month follow-up FGDs with participants to learn about their scale-up experience. Overall, four FGDs were conducted with 8–12 adult ($\geq$ 18 years) participants per FGD, coming to a total of n = 42 (Table 1). The interview guide explored their experience during the TOT training, their scale-up approaches, and their experience implementing the training program. All FGDs were conducted in English, Zulu, and Sotho and audio recorded. All audio recordings were transcribed verbatim for analysis.

**Analytical variables.** We collected socio-demographic data of lay counsellors' who participated in the pilot training, including sex, age, marital status, and highest education level. We assessed household characteristics, including the primary residence location, type of residence, and household members. In addition, we collected participants' English literacy and training background, as well as when they last attended an HIV counselling and testing-related training and how long the training session lasted. In addition, we asked participants when they last tested for HIV and their latest results.

We used a previously described household amenities index, including participants' household characteristics (type of toilet facilities, energy used for cooking, housing structure, household density, and food availability) and ownership of household assets (television, radio, refrigerator, satellite television, cellular telephone, landline telephone, microwave oven, and personal computer) [47]. Total scores for household amenities ranged from 0 to 1, with higher total scores reflecting greater household access to amenities (Cronbach's alpha = 0.81). A cut-off score of 0.3 or less indicated a "low" amenities score, while 0.3 to 0.67 indicated a "medium" amenities score, and a score higher than 0.67 indicated a "high" amenities score.

**MI process and data.** During the pilot training, all roleplay MI sessions involving lay counsellors were audio-recorded with the counsellors' permission. These recordings were utilized to model effective counselling behaviour. Additionally, counsellors submitted audio recordings of counselling sessions with consenting patients after each quarterly contact session. Audio recordings were assessed for fidelity to the MI process using the MITI coding system [42, 43]. MITI includes the following two components: (1) global scores are assigned on a five-point Linkert scale to characterize entire interactions as cultivating change talks (CCT), softening sustain talk (SST), empathy (EM), or partnership (PA), and (2) behaviour counts that require that assessors tally instances of particular counsellor behaviours such as complex

reflections (CR), simple reflections (SR), affirmations, and emphasizing client autonomy [37, 42]. Two independent assessors rated each counsellor. The ratings for each session/participant were averaged to obtain the final values.

**MITI training and supervision.**    The study PI initially trained the MITI assessors to use the MITI system in a two-day workshop. The workshop included didactic instruction, a review of the MITI manual, and group practice using practice examples from the MINT website (https://motivationalinterviewing.org/motivational-interviewing-resources). They also underwent refresher training, which included group practice using a sample of real-life example recordings from the pilot study.

## Outcome measures and analysis

**MI competency.**    Overall, MI adherence scores for each trainee were determined using the following formulae for each MI proficiency category [43]:

- <u>Technical Global (Technical)</u> = (Cultivating Change Talk + Softening Sustain Talk) / 2

- <u>Relational Global (Relational)</u> = (Partnership + Empathy) / 2

We compared the mean scores of each MI adherent measure for each timeline pair using paired t-tests to assess changes in MI proficiency. An overall assessment was conducted using data from the first and last quarters of the training to assess mean differences across time points.

## Data analysis

**Quantitative data analysis.**    Participant characteristics were described using proportions, frequencies, means with standard deviation, and medians with interquartile ranges (IQR), as appropriate. Fisher's exact test was used for discrete variables when expected frequencies were less than five. We conducted a within-intervention group before and after analysis, comparing median scores for skills components at baseline, five months, and 11 months after first contact to determine changes in MI skills as measured by MITI codes. Comparisons were made using the t-test for dependent samples, comparing skill levels before and after training sessions. Data was analyzed using STATA version 14 (StataCorp, College Station, TX).

**Qualitative data analysis.**    All transcripts from the FGDs were analyzed thematically. Initial themes were drawn from topics covered in the interview guide. We also used the HIV treatment readiness framework, the process model for ART readiness, specifically developed for the Thusa-Thuso training program, (S1 Fig) [41], as an additional thematic guide for the analysis. The model acted as an extra thematic guide, centering on crucial aspects such as knowledge, motivation, and readiness for change [41]. We consolidated individual coding through a series of group workshops. Any coder variation was resolved through discussion and consensus from all research team members.

**Ethical review.**    All participants provided written informed consent administered by trained study staff before all data collection procedures. Confidentiality and anonymity were safeguarded by removing all identifiers, including participants' names and names of facilities, from the data. This study was reviewed and approved by the Human Research Ethics Committee (Medical) of the University of the Witwatersrand (Wits HREC M170579).

## Results

### Baseline demographic characteristics of pilot participants

Of the 27 lay counsellors screened during the study period, 100% were eligible, and 25 consented to participate and completed the baseline study interview (Fig 1). Of these, 19 attended

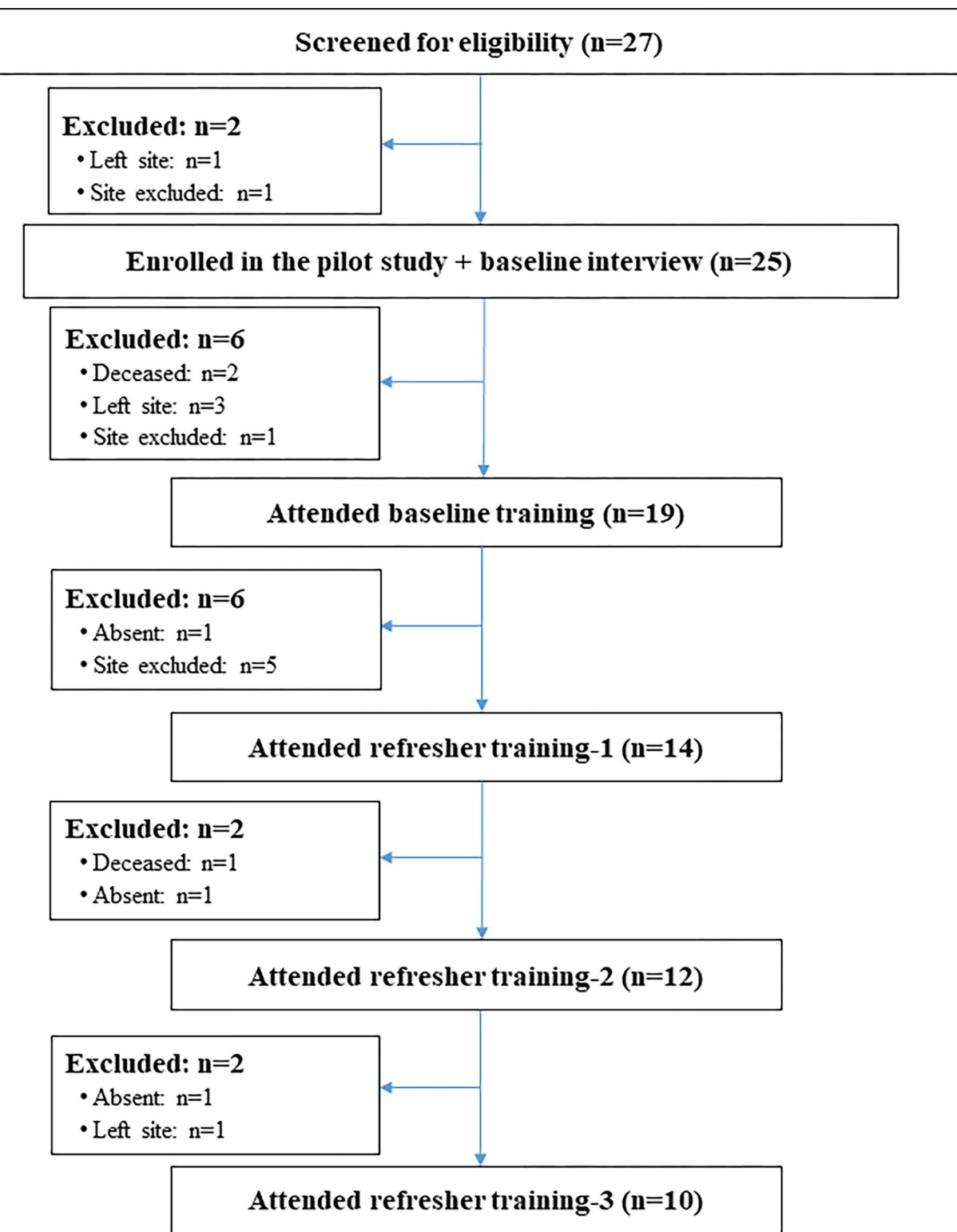

**Fig 1. Recruitment, participant eligibility, and enrolment among adult (≥ 18 years) lay counsellors working in 10 PHC clinics in Johannesburg, South Africa.**

the baseline training; two died, and three were NGO counsellors who were no longer funded and had stopped supporting the site before the baseline training program started. For logistical reasons, two sites were excluded from the pilot training and study. Training support and mentoring were not possible at these clinics because of limitations in infrastructure. Overall, 10 of the original 25 counsellors (40%) completed the training program and participated in the baseline and all refresher training sessions.

We compared baseline socio-demographic characteristics of study participants who completed the training program with those who did not (Table 2). The median age for the cohort was 35 years (IQR: 31–416), with 90.0% being female. All study participants had their most recent HIV test less than six months before study enrolment, and 90.0% reported a negative HIV test result, while one declined to disclose their latest HIV test results. Those who did not complete the training program had similar socio-demographic characteristics to those who did complete the training, except for the type of house and HIV testing history.

Although all counsellors reported high English literacy levels, 40.0% did not complete high school. However, all participants had received basic counsellor training, with 40% having attended the training more than two years prior, and for the majority (90.0%), training lasted for ten days or longer. Additionally, 80% of the counsellors had five years or more experience working as lay counsellors and 60% had spent the duration of their counselling experience working in their current facility. Regarding mental health, one counsellor (10%) screened positive for major depression, 40.0% reported a low level of psychosocial well-being, and 70% reported high levels of perceived social support.

### MI competency scores

MI competencies improved throughout the training program (Fig 2). Technical skills increased from a mean of 2.5 (SD: 0.8) to 3.1 (SD: 0.5), with a mean difference of 0.6 (CI: 0.04, 0.9). The MI relational skills improved from a mean of 3.2 (SD: 0.7) to 3.5 (SD: 0.6), with a mean difference of 0.3 (95% CI: -0.3, 8.5).

### Feasibility, acceptability, and experience of the pilot training among DoH lay counsellors

A total of 11 Department of Health (DoH) lay counsellors (10 female, one male) who attended the pilot training program participated in the FGDs. Overall, lay counsellors were enthusiastic about the training program and appreciated the ongoing support. The analysis of the qualitative counsellor feedback was based on the training process model of readiness as follows:

### Knowledge: Re-orientation of critical basic counselling skills and patient-centred counselling

The counsellors appreciated the refresher and the emphasis on basic counselling knowledge and skills. The approaches used in the training affirmed their baseline counselling knowledge and experience and highlighted areas where the practical application of the conceptual knowledge had been challenging. As a result, counsellors reported improved confidence in their counselling conceptual knowledge, leading to enhanced self-efficacy in their counselling skills.

> "... we have gained like it was an eye-opener also for us to remind ourselves what counselling was all about and the skills which we are supposed to implement every day on counselling in our facilities".–Female, DOH lay counsellor–pilot training participant

**Table 2. Baseline characteristics of lay counsellors who completed the pilot training program compared to those who did not (n = 25).**

| | Training not complete | Training completed | Total | p-value |
|---|---|---|---|---|
| | *No. (col %)* | *No. (col %)* | *No. (col %)* | |
| **Sex** | | | | |
| Female | 15 (100.0) | 9 (90.0) | 24 (96.0) | 0.400 |
| **Age MED (IQR)** | 39 (34–45) | 35 (31–46) | 38 (32–45) | |
| <40 years | 8 (53.3) | 7 (70.0) | 15 (60.0) | 0.678 |
| ≥40 years | 7 (46.7) | 3 (30.0) | 10 (40.0) | |
| **Marital status** | | | | |
| Married | 4 (26.7) | 1 (10.0) | 5 (20.0) | 0.102 |
| In a relationship | 7 (46.7) | 9 (90.0) | 16 (64.0) | |
| Single, no partner | 4 (26.7) | - | 4 (16.0) | |
| **Primary residence** | | | | |
| Current house | 12 (92.3) | 6 (66.7) | 18 (81.8) | 0.264 |
| Another province/rural area | 1 (7.7) | 3 (33.3) | 4 (18.2) | |
| **Lives with** | | | | |
| With partner/spouse | 8 (57.1) | 5 (50.0) | 13 (54.2) | 0.253 |
| Parents/relatives | 4 (28.6) | 1 (10.0) | 5 (20.8) | |
| Alone/with children | 2 (14.3) | 4 (40.0) | 6 (25.0) | |
| **Type of house** | | | | |
| House/ brick structure in its own stand or yard | 13 (92.9) | 4 (40.0) | 17 (70.8) | 0.009 |
| House/ flat/ room in someone's house/ yard or informal dwelling/ shack | 1 (7.1) | 6 (60.0) | 7 (29.2) | |
| **Amenities index** | | | | |
| Low to medium | 4 (33.3) | 6 (75.0) | 10 (50.0) | 0.170 |
| High | 8 (66.7) | 2 (25.0) | 10 (50.0) | |
| **HIV testing history** | | | | |
| Less than 6 months ago | 8 (53.3) | 10 (100.0) | 18 (72.0) | 0.020 |
| 6 months ago, or longer | 7 (46.7) | - | 7 (28.0) | |
| **HIV status** | | | | |
| HIV negative | 8 (53.3) | 9 (90.0) | 17 (68.0) | 0.137 |
| HIV positive | 5 (33.3) | - | 5 (20.0) | |
| Declined to disclose | 2 (13.3) | 1 (10.0) | 3 (12.0) | |
| **Psychological well-being** | | | | |
| Low | 2 (13.3) | 4 (40.0) | 6 (24.0) | 0.175 |
| Moderate to high | 13 (86.7) | 6 (60.0) | 19 (76.0) | |
| **Perceived Social support** | | | | |
| Medium | 3 (20.0) | 3 (30.0) | 6 (24.0) | 0.653 |
| High | 12 (80.0) | 7 (70.0) | 19 (76.0) | |
| **Depression** | | | | |
| No depression | 8 (53.3) | 9 (90.0) | 17 (68.0) | 0.158 |
| low to med depression | 4 (26.7) | - | 4 (16.0) | |
| Major depression | 3 (20.0) | 1 (10.0) | 4 (16.0) | |
| **Highest education level** | | | | |
| High school | 6 (40.0) | 4 (40.0) | 10 (40.0) | 0.089 |
| Completed Grade 12 | 4 (26.7) | 6 (60.0) | 10 (40.0) | |
| >Grade 12 | 5 (33.3) | - | 5 (20.0) | |
| **English literacy** | | | | |
| I can read very well | 10 (66.7) | 10 (100.0) | 20 (80.0) | 0.061 |
| I can read somewhat | 5 (33.3) | - | 5 (20.0) | |

*(Continued)*

**Table 2.** (Continued)

|  | Training not complete | Training completed | Total | p-value |
|---|---|---|---|---|
|  | *No. (col %)* | *No. (col %)* | *No. (col %)* |  |
| **Basic counsellor training attendance** |  |  |  |  |
| ≤24 months ago | 8 (57.1) | 4 (40.0) | 12 (50.0) | 0.680 |
| >24 months ago | 6 (42.9) | 6 (60.0) | 12 (50.0) |  |
| **Basic counsellor training duration** |  |  |  |  |
| <10 days | 1 (6.7) | 1 (10.0) | 2 (8.0) | 1.000 |
| ≥10 days | 14 (93.3) | 9 (90.0) | 23 (92.0) |  |
| **Experience working as a counsellor** |  |  |  |  |
| <5 years | 5 (33.3) | 2 (20.0) | 7 (28.0) | 0.659 |
| ≥5 years | 10 (66.7) | 8 (80.0) | 18 (72.0) |  |
| **Duration working at the current facility** |  |  |  |  |
| <5 years | 7 (46.7) | 4 (40.0) | 11 (44.0) | 1.000 |
| ≥5 years | 8 (53.3) | 6 (60.0) | 14 (56.0) |  |

MED, median; IQR, interquartile range; HIV, human immunodeficiency

Participants highlighted how the MI training helped them re-focus on a patient-centred counselling approach, thus structuring their approaches to building counsellor-client rapport. Several reported that pressure from management to achieve specific HIV testing targets resulted in rushed counselling sessions. MI counselling skills helped them structure the 15–30 minutes they spent with each patient, utilizing MI elicitation techniques to ensure clients felt listened to and supported.

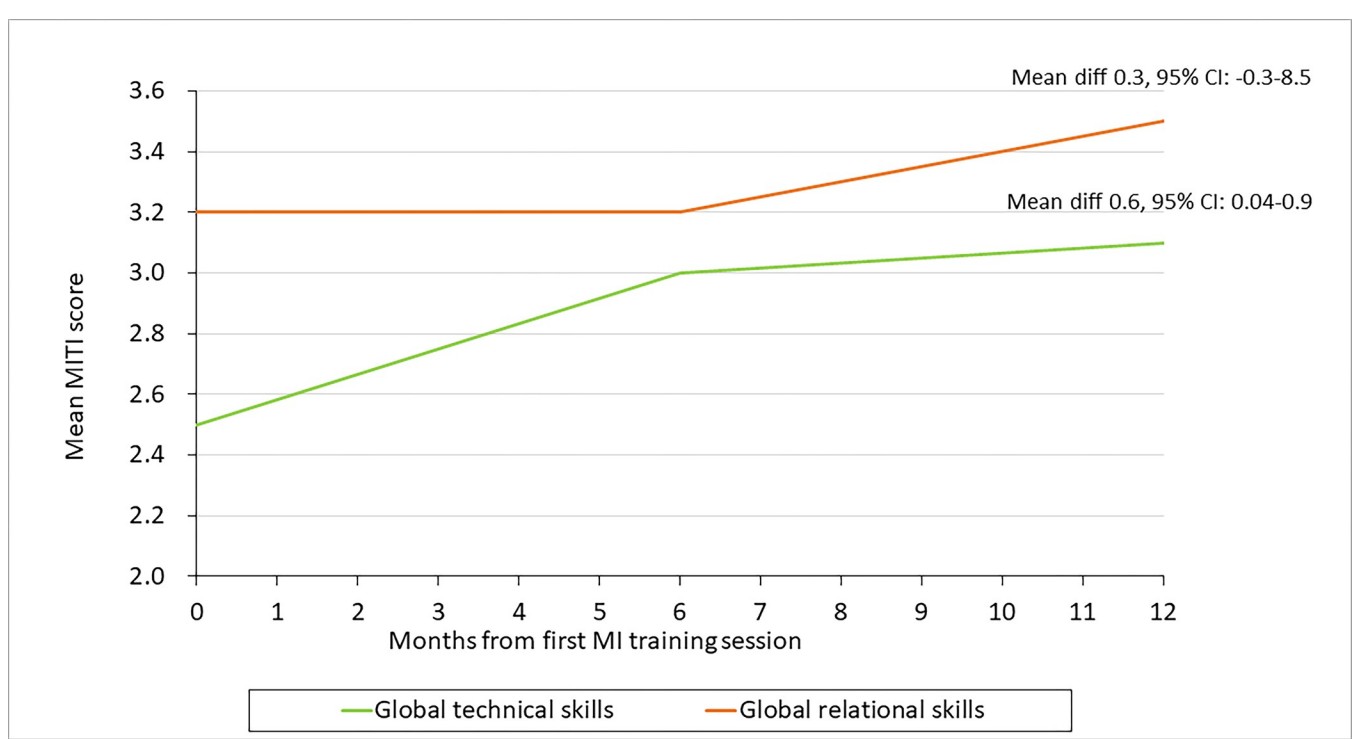

**Fig 2. Global MI technical and relational scores over the 12-month Thusa-Thuso training and support period.**

*". . .before, when we would just test patients, it would just test and go home. We would look at the queue and how long it is and don't check if the patient is satisfied or not. Now I can listen to them and make sure that when they leave, they are satisfied, and everything is right "– Female, DOH lay counsellor–pilot training participant.*

## Motivation: MI counselling skills and perceived importance

The developed training program emphasized the high value of lay counsellors in the care of PLHIV. Therefore, despite experiences of insensitivity within clinic settings, participating counsellors felt a renewed sense of pride in their role in clients' lives and communities. Additionally, participants reported increased self-awareness and mentioned how they had become conscious of how they approached each counselling session (both good methods and errors). Knowing their limitations meant they could ask for assistance from fellow counsellors to support them in difficult situations.

*"So, because he [the clinic manager] said we were going to school [the training], something that made me feel like I was serious. . .I gained a lot of knowledge. I took the training seriously"–Female, DOH lay counsellor–pilot training.*

*"What I like now is that no one will be on your back looking for stats. I know I am not here for the stats; I'm here for people's lives. . ."–Female, DOH lay counsellor–pilot training participant*

*". . .if I feel that the patient is above me too much, I refer to my colleague. I would rather call her to come and attend to them, and they go away feeling right. . ."–Female, DOH lay counsellor–pilot training participant*

## Readiness: In-training guided practice, modelling, onsite practice, and refresher training

The developed training program included practical sessions to prepare counsellors to apply the acquired MI skills in real-work situations. Counsellors could identify with the model counsellor, patients, and scenarios depicted in training modelling videos. They referred to the videos when highlighting examples of correct and incorrect MI counselling behaviours.

Initially, counsellors were nervous about audio recording their sessions for skills review. However, after several sessions, they became more comfortable and could apply the skills they had learned during training. Listening to their counselling session was a new experience for them, and it provided a great opportunity to reflect on their counselling methods and use of MI counselling skills. Through this process, they identified their strengths and areas for improvement. They also valued feedback from the peer reviews of counselling audios.

*"I think, for me, the recordings, the time we were recording ourselves because we were listening to ourselves, that's whereby we see our mistakes. . . So, the recordings I see have worked very well for me."–Female, DOH lay counsellor–pilot training participant.*

Participants viewed the refresher training sessions as valuable, worthwhile, and helpful in clarifying their MI skills. In addition, though counsellors were initially nervous about being observed during counselling sessions, they came to value the mentoring support. They also appreciated how facilitators affirmed their efforts and provided constructive corrective feedback.

*"Then refreshers, I think they helped us a lot too. Unlike maybe if they had just done that first training, we would have forgotten a lot of things. Some of the [concepts] we were hearing them for the first time".–Female, DOH lay counsellor–pilot training participant*

## Scalability following dissemination and early implementation among district support partners

A total of 127 staff members from five NGOs attended the MI TOT workshops between November 2020 and January 2021. 42 TOT participants participated in the FGDs to provide feedback on their training and dissemination experiences (Table 1). Most participants were managers overseeing HTS programs, skills trainers, or supervisors of lay counsellors with extensive counselling skills. However, most had not previously encountered MI counselling. The participants appreciated how the training introduced MI counselling by linking it to their prior basic counselling skills and knowledge. They valued how the MI training prompted them to re-focus on patient-centred counselling. Many mentioned how the HIV program deliverables, including targets, often overshadowed patient needs, increasing the risk of poor-quality service and suboptimal patient experience.

*". . .my overall experience about MI training is that I saw it brought back to the patient-centred approach. . .it brought back the importance of, you know, the patient knowing and actually wanting to take the medication because taking the medication will actually benefit her or him"–Female, NGO staff, TOT participant.*

One of the district support partners involved in the TOT experienced organizational restructuring, which saw some staff members retrenched, including some who participated in the TOT. The remaining participants expressed concerns regarding uncertainty around their continued employment, affecting their motivation for their job and disseminating the Thusa-Thuso MI training program.

*"Well, on my side, I didn't see the importance of that training. . . I wondered why they were taking us to train, knowing we were about to lose our jobs. It was just that even my mind was not there at all"–Male, NGO staff, TOT participant.*

The training language was a barrier that TOT participants had to overcome when implementing what they had learned. Participants had to translate key MI terms into local languages to facilitate understanding MI concepts, which became easier as they conducted more training. Participants perceived the training videos as helpful as they modelled realistic scenarios and were in local languages. The videos also gauged their trainees' understanding of MI concepts and skills during in-training roleplays and in-clinic mentoring support sessions. Trainees indicated they found the competency assessments valuable for continuing on-the-job development and quality assurance. However, TOT participants perceived the MITI review process and tool as too cumbersome for large-scale applications. Hence, some adapted the MITI tool into a scoring form for sit-in mentoring.

*". . . it was best done using the local language because the participants were able to express themselves towards the clients and to dig more because it is easier when it is done in your local language. . ."–Female, NGO staff, TOT participant.*

*"So, incorporate the videos, sort of like more videos to explain some of the concepts, topics such as affirmation, so instead of just it being a lecture, lecture, lecture, let's cover some practical*

*stuff, the videos and the interactive stuff to sort of get the information in better".–Male, NGO staff, TOT participant.*

*"I thought, what made the job easier was also the role plays that were there because we were able to incorporate our daily work into it".–Female, NGO staff, TOT participant.*

## Discussion

Our findings indicate that MI skills can be effectively transferred to lay counsellors using a combination of training methods, including didactic presentations, open discussions, training videos, roleplays, and mentoring, all through simultaneous English and local language translations by bilingual trainers. The participating counsellor's fidelity to the intervention was moderate, suggesting the effectiveness of training procedures, quality assurance through MITI assessments, and mentoring by training facilitators. In addition, it was encouraging to observe the significant improvement in lay counsellors' MI relational skills, particularly in "Empathy and Partnerships", which are key components of MI counselling [23, 26]. In our pilot study cohort, MI global ratings were higher than previous MI-based counselling interventions implemented in similar settings [32, 48–50]. Although our sample size was smaller than other studies, most previous cohorts consisted of shorter training sessions with limited ongoing support. However, it would be difficult to make conclusive comparisons without adequately powered randomized controlled trials.

As MI skills have been reported to fluctuate over time, MI training programs must provide refresher training and ongoing mentoring support [29]. The strengths of the Thusa-Thuso MI training program lie in the fact that the training duration is long enough to allow counsellors to grasp MI concepts and includes quarterly refresher training [26, 29, 36, 51]. Notably, Evangeli et al. (2011) found marked improvements in MI skills when trainees attended follow-up training compared to those associated with the initial one-day training workshop [52]. In qualitative discussions in our study, lay counsellors noted that ongoing support through quarterly refresher training and mentoring enabled them to clarify MI skills learned at the baseline training.

The MITI tool [43] proved an effective quality assurance tool in the training program. Specifically, the tool supported trainers in providing standardized feedback to trainees on MI fidelity. As trainees shared their feedback with the group and trainers, they were able to demonstrate their understanding of key MI concepts and behaviours, which contributed to the overall learning of key MI concepts and behaviours. Previous reviews found that not all MI training programs used validated MI treatment integrity instruments to assess counsellor fidelity [53]. Consequently, it would be difficult to attribute the failure or success of MI interventions to fidelity or the intervention itself [53]. Therefore, the use of MI in implementation should include periodic assessments of treatment integrity using a validated instrument such as the MITI coding tool [43, 53].

However, the MITI tool proved too onerous for large-scale use by NGOs. Instead, partners adapted the tool into a scoring form to document sit-in mentoring sessions for trainees during implementation. Thus, the MITI tool needs to be validated for broader application to organizations scaling up MI counselling for their health programs.

In our broader study, high attrition rates from the year-long training program contributed to post-training skill erosion. For the pilot lay counsellor training, attrition resulted from logistical barriers in implementation and lay counsellor deaths. All deaths occurred among HIV-positive counsellors, but the causes are unknown. NGO-supported lay counsellors enrolled for baseline training were no longer working at the study site because funding and support had

ceased. Additionally, scale-up was also marred by attrition among TOT trainees owing to staff retrenchments related to the cessations of donor funding among NGO partners. Thus, these skills were lost to the organization and NGO-supported public health facilities. The sustainability of acquired skills is critical for supporting health programs to benefit from the scale-up of MI counselling implementation.

### Limitations

A major limitation of this study is the small sample size of the pilot test, attrition caused by staff turnover, and study site challenges, which reduced the sample size even further. In addition, the lack of a control group limits conclusions about the causes of observed changes in MI skill proficiency.

An additional limitation is that participants in the pilot training were from clinics in urban settings. Therefore, application in rural settings remains untested. Furthermore, as participating lay counsellors were aware of the purpose of the study, their counselling recordings may have been influenced by perceived pressure to provide a best-case picture of practice in work and training settings. Moreover, even though participation in the training was not mandatory, lay counsellors were informed by their supervisors of the training program. Thus, there may have been undue expectations that participation in the training could lead to better job opportunities or integration into the formal health workforce. These factors may have initially influenced participants' motivation and enthusiasm for the training.

Despite these limitations, results from the study provide direction and considerations for future, more rigorous studies.

### Conclusions

Our findings suggest participating lay counsellors perceived the Thusa-Thuso MI program as feasible and acceptable. Additionally, scale-up through TOT is possible with flexibility for adaptations without eliminating core MI concepts and skills and applying theory and evidence-based training methodologies. However, larger studies are needed to determine the effect of the Thusa-Thuso program on patient ART adherence and retention.

### Supporting information

**S1 Fig. HIV ART readiness framework.**
(TIF)

### Acknowledgments

We would like to express our gratitude and recognition to the lay counsellors who took part in the pilot training and study, as well as our appreciation for the contributions of Michael Mothapo, who served as the study interviewer, and the dedicated trainers involved in the TOT scale-up initiatives. We also gratefully acknowledge individuals and partner organisations involved at various stages of the implementation of the Thusa-Thuso program pilot study. Right to Care (RTC), ANOVA Health Institute (ANOVA), Wits Reproductive Health Institute (WRHI) and Maternal, Adolescent and Child Health (MatCH); local NGO partners to the South African National and Gauteng provincial Department of Health, for participating in the scale-up strategy for the training program. We are also grateful to clinic managers from clinics in the Johannesburg Health District and district staff, including district managers, for supporting the implementation of the pilot training.

## Author Contributions

**Conceptualization:** Dorina Onoya.

**Data curation:** Idah Mokhele, Tembeka Sineke, Dorina Onoya.

**Formal analysis:** Idah Mokhele, Dorina Onoya.

**Funding acquisition:** Dorina Onoya.

**Investigation:** Dorina Onoya.

**Methodology:** Dorina Onoya.

**Project administration:** Idah Mokhele, Tembeka Sineke, Dorina Onoya.

**Resources:** Dorina Onoya.

**Supervision:** Robert A. C. Ruiter.

**Writing – original draft:** Idah Mokhele, Dorina Onoya.

**Writing – review & editing:** Idah Mokhele, Tembeka Sineke, Marnie Vujovic, Robert A. C. Ruiter, Jacqui Miot, Dorina Onoya.

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
