## [Decision Letter · Decision Letter 0]

12 Dec 2023

PGPH-D-23-01878

Improving patient-centred counselling skills among lay healthcare workers in South Africa using the Thusa-Thuso motivational interviewing training and support program

Dear Dr. Mokhele,

Thank you for submitting your manuscript to PLOS Global Public Health. After careful consideration, we feel that it has merit but does not fully meet PLOS Global Public Health’s publication criteria as it currently stands. Therefore, we invite you to submit a revised version of the manuscript that addresses the points raised during the review process.

Please ensure that your decision is justified on PLOS Global Public Health’s publication criteria and not, for example, on novelty or perceived impact.

We look forward to receiving your revised manuscript.

Kind regards,

Nnodimele Onuigbo Atulomah, PhD

Academic Editor

Journal Requirements:

b. If any authors received a salary from any of your funders, please state which authors and which funders.

3. Please provide separate figure files in .tif or .eps format only and remove any figures embedded in your manuscript file. Please also ensure all files are under our size limit of 10MB.

4. In the online submission form, you indicated that "The data that support the findings of this study are available from the corresponding author, DO, upon reasonable request". All PLOS journals now require all data underlying the findings described in their manuscript to be freely available to other researchers, either 1. In a public repository, 2. Within the manuscript itself, or 3. Uploaded as supplementary information.

Additional Editor Comments (if provided):

The manuscript has merits for publication considerations but requires editorial cleaning. Kindly go through the manuscript thoroughly for grammatical corrections or submit to special language editing service as suggested by one of the reviewers.

Reviewers' comments:

Reviewer's Responses to Questions

**Comments to the Author**

1. Does this manuscript meet PLOS Global Public Health’s publication criteria? Is the manuscript technically sound, and do the data support the conclusions? The manuscript must describe methodologically and ethically rigorous research with conclusions that are appropriately drawn based on the data presented.

Reviewer #1: Yes

Reviewer #2: Yes

2. Has the statistical analysis been performed appropriately and rigorously?

Reviewer #1: Yes

Reviewer #2: Yes

3. Have the authors made all data underlying the findings in their manuscript fully available (please refer to the Data Availability Statement at the start of the manuscript PDF file)?

Reviewer #1: Yes

Reviewer #2: Yes

4. Is the manuscript presented in an intelligible fashion and written in standard English?

Reviewer #1: Yes

Reviewer #2: Yes

5. Review Comments to the Author

Reviewer #1: The authors painstakingly researched and presented their finding in their manuscript. Effort was directed at utilizing both qualitative and quantitative data evaluation techniques, and the limitations of the study were clearly spelt out. The only a few participants eventually completed the 12-month motivational Interviewing (MI) training and support program for the lay counsellors.

Reviewer #2: Replicability

The methods, data collection and data analysis can be reproduced.

Statistical review

Applicable

Scholarly contributions

This study contributes to the global literature on HIV based interventions in the global south.

The strengths and limitations of the study

Strengths

• The combined use of qualitative and quantitative research approaches.

Limitations

• None.

The manuscript should be double-checked again for grammar mistakes.

6. PLOS authors have the option to publish the peer review history of their article (what does this mean?). If published, this will include your full peer review and any attached files.

**Do you want your identity to be public for this peer review?** For information about this choice, including consent withdrawal, please see our Privacy Policy.

Reviewer #1: No

Reviewer #2: **Yes: **Saheed Akinmayowa Lawal

---

## [Editor Report · Decision Letter 1]

21 Mar 2024

Improving patient-centred counselling skills among lay healthcare workers in South Africa using the Thusa-Thuso motivational interviewing training and support program

PGPH-D-23-01878R1

Dear Dr. Onoya,

We are pleased to inform you that your manuscript 'Improving patient-centred counselling skills among lay healthcare workers in South Africa using the Thusa-Thuso motivational interviewing training and support program' has been provisionally accepted for publication in PLOS Global Public Health.

Best regards,

Nnodimele Onuigbo Atulomah, PhD

Academic Editor

Having reviewed the revised manuscript along with the comments of the reviewers and observing that the authors have adhered strictly to address all raised issues and revise all comment requiring amendments, I am inclined recommend that this manuscript be moved to the next level for publication. All corrections have been faithfully made to the satisfaction of the Academic Editor.